# Characterization and Identification of Prenylated Flavonoids from *Artocarpus heterophyllus* Lam. Roots by Quadrupole Time-Of-Flight and Linear Trap Quadrupole Orbitrap Mass Spectrometry

**DOI:** 10.3390/molecules24244591

**Published:** 2019-12-15

**Authors:** Jin-Bao Ye, Gang Ren, Wen-Yan Li, Guo-Yue Zhong, Min Zhang, Jin-Bin Yuan, Ting Lu

**Affiliations:** 1Research Center of Natural Resources of Chinese Medicinal Materials and Ethnic Medicine, Jiangxi University of Traditional Chinese Medicine, Nanchang 330004, China; 201781800024@jxutcm.edu.cn (J.-B.Y.); liwenyan369@126.com (W.-Y.L.); zgy1037@163.com (G.-Y.Z.); 18770839821@163.com (T.L.); 2Key Laboratory of Modern Preparation of TCM, Ministry of Education, Jiangxi University of Traditional Chinese Medicine, Nanchang 330004, China; zfishm@163.com

**Keywords:** prenylated flavonoids, *Artocarpus heterophyllus*, Q-TOF-MS, LTQ-Orbitrap-MS, fragmentation rules, identification

## Abstract

In this study, a combination of quadrupole time-of-flight mass spectrometry (Q-TOF-MS) and linear trap quadrupole orbitrap mass spectrometry (LTQ-Orbitrap-MS) was performed to investigate the fragmentation behaviors of prenylated flavonoids (PFs) from *Artocarpus* plants. Fifteen PFs were selected as the model molecules and divided into five types (groups A–E) according to their structural characteristics in terms of the position and existing form of prenyl substitution in the flavone skeleton. The LTQ-Orbitrap-MS^n^ spectra of the [M − H]^−^ ions for these compounds provided a wealth of structural information on the five different types of compounds. The main fragmentation pathways of group A were the ortho effect and retro Diels–Alder (RDA), and common losses of C_4_H_10_, CO, and CO_2_. The compounds in group B easily lose C_6_H_12_, forming a stable structure of a 1,4-dienyl group, unlike those in group A. The fragmentation pathway for group C is characterized by obvious ^1,4^A^−^, ^1,4^B^−^ cracking of the C ring. The diagnostic fragmentation for group D is obvious RDA cracking of the C ring and the successive loss of CH_3_ and H_2_O in the LTQ-Orbitrap-MS^n^ spectra. Fragmentation with successive loss of CO or CO_2_, ·CH_3_, and CH_4_ in the LTQ-Orbitrap-MS^n^ spectra formed the characteristics of group E. The summarized fragmentation rules were successfully exploited to identify PFs from *Artocarpus*
*heterophyllus*, a well-known *Artocarpus* plant, which led to the identification of a total of 47 PFs in this plant.

## 1. Introduction

Prenylated flavonoids (PFs) are a class of structurally distinct chromone derivatives characterized by several prenyl units linked to a flavone nucleus by C–C and/or C–O bonds. PFs possess high structural diversity due to various prenyl substitution patterns on the flavone skeleton, which correspond to their wide range of biological activities such as cytotoxicity [1], tyrosinase inhibition [2], and cathepsin K inhibition [3,4], as well as antiplasmodial [5], antitrypanosomal [5], and anti-inflammatory [6] activity, and inhibition of neutrophil respiratory burst [7]. Due to their complex chemical diversity and wide range of biological activities, the search for structurally interesting and biologically active PFs from natural products has attracted the intense attention of natural product chemists and pharmacologists in the past few decades. Interestingly, the occurrence of PFs appears to have a significantly chemotaxonomical tendency and is limited to within only a few plant families, including Moraceae. The genus *Artocarpus* of Moraceae has been reported as a rich source of PFs, and more than 300 have been isolated from *Artocarpus* plants to date [8,9].

*Artocarpus heterophyllus* Lam., a well-known member of *Artocarpus*, is an evergreen arbor widely distributed in tropical and subtropical regions of Asia, such as Thailand, Vietnam, the Philippines, and other countries [10]. In China, *A. heterophyllus* is popularly cultivated for its popular edible fruits, while its root has been used in folk medicine for subduing swelling and detoxicating [11]. A variety of bioactive PFs were previously identified from *A. heterophyllus* using the conventional process of isolation and purification on a preparative scale followed by structure elucidation via spectroscopic methods [4,12,13]. However, the purification of PFs from the crude extract by this conventional method is usually time-consuming, tedious, and labor-intensive due to their thermal sensitivity and low abundance in *A*. *heterophyllus*.

The combination of quadrupole time-of-flight mass spectrometry (Q-TOF-MS) and linear trap quadrupole orbitrap mass spectrometry (LTQ-Orbitrap-MS) is a powerful analytical strategy to rapidly screen and identify trace compounds. The combined method possesses high resolution, high mass accuracy, high sensitivity, and abundance fragment information. It is frequently used for the structural characterization of natural products in crude extracts, especially when the reference compounds are unavailable [14,15,16]. Despite there being vast numbers of known PFs, their fragmentation behaviors and corresponding mechanisms remain relatively unexplored.

In the present work, 15 PFs previously isolated and identified [17,18,19,20] from the extract of *Artocarpus* plants were selected as model molecules and divided into five types according to the substitution pattern of prenylated groups in the flavone nucleus. The fragmentation behaviors of all the five types of PFs were systematically investigated by Q-TOF-MS and LTQ-Orbitrap-MS^n^. Based on these summarized fragmentation rules, a total of 47 trace PFs were identified. In this work we developed an effective mass spectrometric method for the rapid identification of PFs in mixtures; this could also provide a valuable aid for the characterization of PFs in other PF-containing medicinal herbs.

## 2. Results and Discussion

The chemical structures of the 15 reference PFs are shown in Figure 1. According to the positions and existence of prenylated substitution in the flavone skeleton, these compounds fall into five categories: group A: free 6- or 8-prenylated substituted flavones (PFs **1**–**3**); group B: free 8-geranyl substituted flavones (PFs **4** and **5**); group C: free 3-prenylated substituted flavones (PFs **6**–**9**); group D: 3-partial heterocyclization-prenoid substituted flavones (PFs **10** and **11**); and group E: dihydrobenzoxanthone derivatives (PFs **12**–**15**).

All the investigated reference compounds were initially analyzed by electrospray ionization quadrupole time-of-flight (ESI-Q-TOF) mass spectrometry in both positive and negative ion modes. Each group of PFs exhibited a strong molecular ion peak ([M − H]^−^ in negative ion mode) with a very small percentage of fragments in the Q-TOF-MS spectrum, and yielded a wealth of product ions in the subsequent TOF-MS/MS and LTQ-Orbitrap-MS^n^ spectra. Accordingly, the [M − H]^−^ ions were selected as the precursor ions for collision-induced dissociation (CID) fragmentation to produce MS/MS spectra, and the relatively prominent product ions were then chosen for further MS^n^ analysis. As summarized in Table 1, these MS^n^ spectra provided abundant fragment ions for the structural characterization and identification of PFs. Using this rule, a total of 47 PFs in the extract of the *A. heterophyllus* were rapidly identified by ultra-high performance liquid chromatography (UPLC)-Q-TOF-MS/MS.

### 2.1. Fragment Naming Rules

In this study, the C–C bond cleavages of different groups occurred at different positions of the C ring. In general, the fragment ions formed by the C–C bond cleavage on the C ring of flavonoids are named ^i,j^A and ^i,j^B [21], where A and B represent benzene ring A and benzene ring B, respectively, on the parent nucleus of the flavonoid. “i” and “j” represent the fracture position of the C–C bond cleavage on the C ring, as shown in Figure 2.

### 2.2. Fragmentation of PFs ***1***–***3***

PFs **1**–**3** in group A represent a type of general prenylated flavones with coplanar structure and bearing a free prenyl moiety at the C-6- or -8 position. PFs **1–3** mainly showed the neutral loss of C_4_H_8_ or C_5_H_10_ via the ortho effect at the C-8 position to yield a prominent ion at *m*/*z* 297. This fragmentation pathway was found for the first time in PFs. In addition, the product ions ^1,3^A^−^ and/or ^1,3^B^−^ could be deduced, and their fragmentation pathway was retro-Diels–Alder (RDA) cleavage from the 1,3-position of the C-ring. Common losses of other neutral molecules, such as C_4_H_10_, CO, and CO_2_, were also observed.

PF **1** was selected as an example to illustrate the fragmentation characteristics of group A. Figure 3, Appendix A exhibits the typical ESI-MS spectra and fragmentation pathways of PF **1** in negative ion mode by Q-TOF-MS/MS and LTQ-Orbitrap MS^n^ (*n* = 2–4) (Appendix A). The Q-TOF-MS spectrum (Figure 3A) of PF **1** showed a precursor [M − H]^−^ ion at *m*/*z* 367.1205 (error 4.90 ppm). In the Q-TOF-MS/MS spectrum (Figure 3B), the prominent ion at *m*/*z* 297.0412 ([M − H − C_5_H_10_]^−^, error 2.36 ppm) was from the loss of C_5_H_10_ via the ortho effect at the C-8 position, and the fragmentation at *m*/*z* 309.0417 ([M − H − C_4_H_10_]^−^, error 3.88 ppm) was from the loss of C_4_H_10_ at the same position. Simultaneously, a minor ion at *m*/*z* 233.0852 was also detected, owing to RDA fragmentation at the 1,3-position of the C-ring. As shown in Figure 3C, further loss of CO_2_ from the ion at *m*/*z* 233.0852 generated the product ion at *m*/*z* 189.0938 in the LTQ-Orbitrap-MS^3^ spectrum. As seen from the MS^3–4^ spectra (Figure 3D,E), a fragment ion at *m*/*z* 309.0417 was observed with the initial loss of the CO_2_ group at the C ring; subsequently, loss of the CO group at the B ring position yielded prominent ions at *m*/*z* 265.0599 [M − H − C_4_H_10_ − CO_2_]^−^ and *m*/*z* 237.0466 [M − H − C_4_H_10_ − CO_2_ − CO]^−^. In addition, the fragment ion at *m*/*z* 175.0082 from *m*/*z* 309.0417 via the classical RDA cleavage pathway was also observed. Figure 3F summarizes the above fragmentation pathway of PF **1**.

The fragmentation pathways of PFs **1**–**3** are almost identical, which is in full support of the structural characterization of this type of PF. The characteristic fragment ions described above can be regarded as the important information for the qualification of group A. There are still some differences in the cleavage pathways of the prenyl moieties at C-6 (PF **3**) and C-8 (PFs **1** and **2**). The fragment ions of PF **3** are relatively few, and the ortho effect was not observed. PF **3** exhibited RDA cleavage at the C-6 position, resulting in a prominent ion at *m*/*z* 219.0673 ([M − H − ^1,3^B^−^], error 4.57 ppm) and a fragment ion at *m*/*z* 133.0230 ([M − H − ^1,3^A]^−^ error 3.76 ppm). However, only one fragment ion appeared via RDA cleavage for PFs **1** and **2**. These features could help us to differentiate the compounds bearing a free prenoid moiety at the C-6 position from those bearing the same at the C-8 position.

### 2.3. Fragmentation of PFs ***4*** and ***5***

A detailed comparison revealed that the flavones in groups B and A exhibited much structural similarity. The structural difference lies in the side chains at the C-8 position: geranyl for group B, and prenyl for group A. Compared with the fragmentation behavior of group A, the ortho effect and RDA cleavage in group B exhibited similar characteristics. The difference is that the geranyl group of group B easily loses C_6_H_12_, forming a stable structure of a 1,4 dienyl group. 

PF **4** was selected as an example to illustrate the fragmentation pathways of this class of compounds. Figure 4, Appendix A shows the typical ESI-MS spectra and fragmentation pathways of PF **4** in negative ion mode by Q-TOF-MS/MS and LTQ-Orbitrap-MS^n^ (*n* = 2–4). Q-TOF-MS (Figure 4A) of PF **4** gave a precursor [M − H]^−^ ion at *m*/*z* 421.1644 (error 3.09 ppm). In the Q-TOF-MS/MS spectrum, Figure 4B shows that the prominent ion at *m/z* 297.0395 ([M − H − C_9_H_16_]^−^, error 3.37 ppm) could be attributed to the loss of C_9_H_16_ from the precursor at *m*/*z* 421.1644 via the ortho effect. The precursor at *m*/*z* 421.1644 is able to experience RDA cleavage to form *m*/*z* 133.0301 ([M − H − ^1,3^A]^−^, error 4.51 ppm), similarly to group A. In comparison with group A, owing to the presence of the geranyl (instead of prenyl) side chain at the C-8 position, group B shows obvious cleavage of the geranyl group and thus loses a C_6_H_12_ group. This tendency is more obvious when there is a methoxy group in the mother nucleus, such as in PF **5**. Figure 4C,D shows the successive loss of CO_2_ and CO groups at the C ring, yielding a prominent ion at *m*/*z* 225.0474 [M − H − C_9_H_16_ − CO_2_ − CO]^−^ from *m*/*z* 297.0395. In addition, the primary daughter ion [M − H − C_9_H_16_ − CO]^−^ was observed owing to the loss of CO from the product ion at *m*/*z* 297.0395. The fragmentation patterns of PF **5** are almost identical to those of PF **4**, in full support of the structural characterization of this type of flavone. Figure 4E summarizes the fragmentation pathways to help identify these compounds, taking PF **4** as an example.

### 2.4. Fragmentation of PFs ***6***–***9***

Group C contains four free 3-prenylated substituted flavones, with three of them bearing another one or more prenyl moieties at the C-6, -8, and -3′ positions, but not PF **6**. For these PFs in group C, non-coplanarity is the common structural feature and results from the free 3-prenylated group in the flavone nucleus forcing the 2-phenyl group out of plane with the chromone ring. This unique stereo configuration of the free 3-prenylated substituted flavones in group C leads to significantly different fragmentation behaviors from the other four groups. The most prominent difference between group C and the other groups is in the fragmentation of the C ring. The C ring of this kind of compound shows obvious ^1,4^A^−^, ^1,4^B^−^ cracking, while the common flavonoids show RDA cracking (^1,3^A^−^, ^1,3^B^−^).

As shown in Figure 5, Appendix A PF **6** was taken as an example to illustrate the typical cleavage pathways. Figure 5A shows the molecular ion ([M − H]^−^, *m*/*z* 353.1028, error 0.79 ppm) corresponding to the molecular formula C_26_H_28_O_6_. The product ions [M − H − B^1,4^]^−^ at *m*/*z* 125.0242 (error 1.60 ppm) and [M − H − A^1,4^]^−^ at *m*/*z* 231.1021 (error 2.60 ppm) were from the cracking of the C ring (Figure 5B). The other three compounds all have a similar fragmentation pathway. This kind of cracking was observed only in group C, and not in the other four groups. 

PF **6** exhibited a prominent product ion ([M − H − C_5_H_9_]^−^, *m*/*z* 284.0326, error 0 ppm) which was from the loss of C_5_H_9_ from the precursor ion (Figure 5B). In addition, the strong product ion peak at *m*/*z* 151.0038 ([M − H − ^1,3^B]^−^, error 0.62 ppm) was easily obtained via RDA cleavage from the precursor [M − H]^−^. PF **7** has the same cleavage pathway with a very weak peak ([M − H − ^1,3^B]^−^), but PFs **8** and **9** do not have this fragmentation pathway. This characteristic fragmentation pathway can be used to explore the position and number of prenylated groups of these compounds.

Remarkably, these observations indicated that PFs **7**–**9** are apt to lose the C_4_H_8_ group at the free 3-prenylated group, forming a stable furan ring closed with the hydroxyl at the C-2′ position, and to successively eliminate a C_4_H_8_ group at the C-8 position, forming a double bond between C-7 and C-8 via the ortho effect. Taking PF **8** as an example, this cracking pathway eventually produced a prominent ion at *m*/*z* 309.0432 ([M − H − C_8_H_16_]^−^, error 3.88 ppm). The characteristic fragmentation pathway of group C is of great significance for the identification of these compounds containing more than two prenyl groups.

### 2.5. Fragmentation of PFs ***10*** and ***11***

The two compounds in group D are also 3-prenylated flavones similar to those in group C, but they possess a coplanar structure in which the prenyl substituent at the 3-position is partially heterocyclized to form a pyran ring by C–O linkage between C-11 of the prenylated moiety and the hydroxyl group at the C-2′ position of the B ring. The existence of the C-11/C-2′ pyran ring in group D resulted in a wealth of product ions in the MS^n^ (*n* = 2–5) spectra, which is notably different from group C. The most significant difference between group D and C is the cracking behavior of the C ring: RDA cleavage mainly occurred in group D, and ^1,4^A^−^/^1,4^B^−^ cracking occurred in group C. The difference could be used to differentiate the two groups.

PF **10** was selected as an example to illustrate the fragmentation pathways of group D. Figure 6, Appendix A exhibits the typical ESI-MS spectra and fragmentation pathways of PF **10** in negative ion mode by Q-TOF-MS/MS and LTQ-Orbitrap-MS^n^ (*n* = 2–4). The mass spectrum of PF **10** in Figure 6A showed a precursor ion [M − H]^−^ at *m*/*z* 417.1358 ([M − H]^−^, error 3.41 ppm). PF **10** is mainly cracked via RDA, producing the fragment ions at *m*/*z* 217.0516 ([M − H − ^1,3^B]^−^) in the Q-TOF-MS/MS spectrum (Figure 6B). In addition, the fragment ion at *m*/*z* 361.0737 ([M − H − 4·CH_3_]^−^, error 3.60 ppm) came from the loss of ·CH_3_ from the precursor ion [M-H]^-^. In the LTQ-Orbitrap-MS^3–4^ spectra of PF **10** (Figure 6C,D), the product ion at *m*/*z* 343.0523 was from the loss of one H_2_O, and the fragment ion at *m*/*z* 199.0301 was obtained from the subsequent cracking of the C ring. Figure 6E summarizes the fragmentation pathways of PF **10** in the Q-TOF-MS/MS and LTQ-Orbitrap-MS^n^ (*n* = 2–4) results.

It is noteworthy that the compounds containing a free prenylated group at the C-8 position show some different fragmentation pathways. For the compound PF **11**, the main fragment pathway is the loss of CO_2_ from the C ring, generating the product ion at *m*/*z* 457.2002 ([M − H − CO_2_]^−^, error 3.94 ppm). However, for PF **10**, the main pathway is the loss of H_2_O. This difference can help us to determine the number and location of prenylated groups in compounds of this group.

### 2.6. Fragmentation of PFs ***12***–***15***

The four flavones in group E, with a dihydrobenzoxanthone skeleton, represent the characteristic constituents of the *Artocarpus* species, and this type of flavone has never been found in any other species. It was observed that dihydrobenzoxanthone-type flavone derivatives have a unique structure in which C–C linkage takes place between the C-6′ position of the B ring and the C-12 position of the prenyl moiety located at the C-3 position. Because of this unique structure, the cleavage behavior of this type of flavonoid is different from that of other groups.

PF **12** was selected as an example to illustrate the fragmentation characteristics of this type of compound. Figure 7, Appendix A shows the typical ESI-MS spectra and cleavage pathways of PF **12** in negative ion mode for Q-TOF-MS/MS and LTQ-Orbitrap-MS^n^ (*n* = 2–4). The precursor ion [M − H]^−^ of PF **12** at *m*/*z* 501.1943 (error 4.00 ppm) is shown in Figure 7A. The fragment ions at *m*/*z* 445.1296 ([M − H − 4·CH_3_]^−^, error 0.47 ppm) and *m/z* 485.1626 ([M − H − CH_4_]^−^, error 4.12 ppm) correspond to the loss of 4·CH_3_ or CH_4_ from the molecular ion (Figure 7B). In the LTQ-Orbitrap-MS^3–4^ spectra (Figure 7C,D), we observed the initial loss of the H_2_O group at the C-4’ position from the fragment ion (*m*/*z* 445.1296), and subsequently the loss of the CO group at the C-2’ position, yielding prominent ions at *m*/*z* 399.1040 [M − H − 4·CH_3_ − H_2_O − CO]^−^. In addition, Figure 7E,F illustrates the successive losses of CO and another four ·CH_3_ in the fragmentation [M − H − CH_4_]^−^, producing the product ions [M − H − CH_4_ − CO]^−^ (*m*/*z* 457.1961), [M − H − CH_4_ − 4·CH_3_]^−^ (*m*/*z* 429.1201), and [M − H − CH_4_ − 4CH_3_ − CO]^−^ (*m*/*z* 401.1094). The three other compounds in this group also showed the same fragmentation patterns as PF **12**. Thus, Figure 7G summarizes the possible fragmentation pathway of PF **12**. This fragmentation pathway makes it easier to determine the structural characteristics of these compounds.

### 2.7. Rapid Identification of PFs in the Extract of A. heterophyllus Roots

The PFs in the extract of *A. heterophyllus* roots were analyzed by UPLC-Q-TOF-MS/MS, and the total ion chromatogram (TIC) of the sample in negative ion mode was obtained, as shown in Figure 8A. Based on the existing literature and our previous studies, a total of 47 PFs were identified in the extract of *A. heterophyllus* roots. The formulae of the 47 compounds were accurately assigned within a relative theoretical mass error of 5 ppm, and the relative theoretical mass errors of the characteristic fragment ions were all within 10 ppm. The identification process of the detected compounds was as follows [22]. Firstly, an accurate molecular mass was obtained via the high-resolution Q-TOF-MS technique according to the deprotonated ion [M − H]^−^. Secondly, the formula was obtained by Peakview software according to the accurate molecular mass, constituent elements, and isotope abundance. Thirdly, the differentiation and characterization of the analytes were completed by considering the product ions, fragmentation pathways, and literature data. The related information is summarized in Table 2, including the compound name, molecular formula, precursor ion, characteristic fragment ions, and the error. The identified compounds were validated by extracting the relative characteristic fragmentations, such as precursor ions, nuclear parent ions, and daughter ions (Figure 8). The following examples are given to illustrate the identification process of the PFs.

The precursor ion and fragment ions of peaks 3, 7, 17, 33 and 41 in the TIC profile of *A. heterophyllus* root extract are basically the same as those for PFs **6, 13, 1, 15** and **12** discussed in this experiment. Therefore, compounds **3**, **7**, **17**, **33** and **41** were identified as PFs **6, 13**, **1**, **15** and **12**. The precursor ions of compounds **11** and **28** were *m*/*z* 397.1289 ([M − H]^−^) and *m*/*z* 503.2068 ([M − H]^−^), respectively, affording the molecular formulae C_22_H_22_O_7_ and C_30_H_32_O_7_ with errors of −0.9 and −1.4 ppm, respectively. In the Q-TOF-MS/MS spectra, these two compounds have ^1,4^A^−^, ^1,4^B^−^ fragments. According to the fragmentation pattern of group C, it can be inferred that both of them have a prenylated substitution in the C-3 position. In addition, the fragment ions of compound **10** included [M − H − 2OCH_3_]^−^ (*m*/*z* 335.0935). It can be inferred that compound **11** may contain two OCH_3_. By comparison with our data [4], compound 11 was identified as artoindonesianin Q. Compound **28** also produced a low abundance of RDA fragments. Based on the fragmentation pattern of group C, the C-8 position of compound **28** may contain a prenylated group, but there is no ortho effect. It is possible that there is no free phenolic hydroxyl at the C-7 position. Based on the related literature [17] and retention time, compound 28 was identified as heterophyllin.

Compound **31** produced a 433.1293 [M − H]^−^ precursor ion in negative ion mode. It was inferred that its molecular formula is C_25_H_22_O_7_, and the error was −0.1 ppm. The characteristic fragment ions of compound **31** include [M − H − CO_2_]^−^ (*m*/*z* 389.1372), [M − H − H_2_O]^−^ (*m*/*z* 415.1192), and [M − H − ^1,3^A]^−^ (*m*/*z* 217.0516). It can be inferred that in compound **30**, there is a 3-prenyl side chain forming a pyran ring with HO-6’ of the B ring, and C-3’ and C-4’ of the B ring contain two free phenolic hydroxyl groups. This is basically consistent with the fragmentation pathway of group D. This information suggested that this compound should be 5’-hydroxycudraflavone [17]. The precursor ion of compound **39** ([M − H]^−^ ion at *m*/*z* 503.2086) afforded the molecular formula C_30_H_32_O_7_ with an error of 2.2 ppm. The fragmentations [M − H − C_4_H_8_]^−^ at *m*/*z* 447.1422 and [M – H − CO]^−^ at *m*/*z* 475.2103 indicated that this compound was artonin G [44]. This is basically consistent with the fragmentation pathway of group E. 

## 3. Materials and Methods

### 3.1. Plant Material

The roots of *A. heterophyllus* were collected from Nanning city, Guangxi Zhuang Autonomous Region, China, in April 2017 and authenticated by Associate Researcher Lv Shihong of Guangxi Institute of Botany, Chinese Academy of Sciences. The voucher specimen (TCM20170502) was deposited in the Herbarium of the Department of Pharmacognosy, Research Center of Natural Resources of Chinese Medicinal Materials and Ethnic Medicine, Jiangxi University of Traditional Chinese Medicine, Nanchang, China.

### 3.2. Model Molecules and Reagents

A total of 15 PFs (PFs **1–15**) were isolated from *Artocarpus* plants in our previous studies [17,18,19,20]. Their structures (Figure 1) were unequivocally elucidated by spectroscopic methods, including one-dimensional and two-dimensional nuclear magnetic resonance, high-resolution electrospray ionization mass spectroscopy, and infrared spectroscopy. The purity of these compounds was determined to be higher than 98% by normalizing the peak area using high-performance liquid chromatography (HPLC) equipped with a diode array detector (DAD). Chromatographic-grade acetonitrile (ACN) and methanol were purchased from Tedia Co. Inc. Deionized water was prepared using a Milli-Q water purification system (Millipore, USA).

### 3.3. Sample Preparation

The air-dried and powdered roots of *A. heterophyllus* (50.0 g) were extracted with 95% EtOH three times (500 mL for each extraction) at room temperature. The filtrate was evaporated in vacuo to produce a residue (4.2 g) which was loaded on an HP-20 macroporous resin column chromatograph and eluted successively with a gradient of EtOH/H_2_O (0:100 and 95:5, *v*/*v*) to give two fractions. The 95% EtOH elution fraction was concentrated to afford a residue. The dried residue was re-dissolved in 50 mL of methanol in a volumetric flask; then, the samples were filtered through a 0.22 μm membrane prior to analysis.

### 3.4. Ultra-High Performance Liquid Chromatography (UPLC)-Q-TOF-MS/MS and UPLC-LTQ-Orbitrap-MS^n^ Analysis of PFs

A TripleTOF AB 5600^+^ high-resolution mass spectrometer (AB SCIEX, Framingham, USA) was used to obtain primary and secondary mass spectrometry data by the UPLC-Q-TOF-MS/MS method. The AB mass spectrometer was equipped with an electrospray ionization (ESI) ion source and an LC-30A ultra-high-performance liquid chromatograph (Shimadzu, Japan). The multistage mass spectrometry data were obtained using a Thermo Fisher LTQ-Orbitrap Electrostatic Mass Spectrometer (Thermo Fisher Scientific, Bremen, Germany).

In the UPLC-Q-TOF/MS analysis, the chromatographic separation was carried out on an ACQUIT UPLC^®^ BEH (1.7 μm, 2.1 × 50 mm, Water, Milford, MA, USA). The mobile phase was composed of 30% ultrapure water and 70% methanol with a flow rate of 0.3 mL/min, and the injection volume was 2.0 μL. The mobile phase for the extracts was ultrapure water (A) and methanol (B) using a gradient elution program: 45–60% B at 0–2 min, 60–65% B at 2–10 min, 65–70% B at 10–25 min, 70–80% B at 25–30 min, 80–100% B at 30–40 min, then held for 10 min. Q-TOF-MS analysis was performed in negative ion mode using full scan mode with a mass range of 100–1500 Da. The MS parameters were optimized as follows: ion spray voltage, −4500 V; source temperature, 550 °C; curtain gas, 30 psi; nebulizer gas (GS1), 50 psi; heater gas (GS2), 50 psi; de-clustering potential (DP), −100 V; and pre-scan and trigger second-stage scan time of Q-TOF-MS, 250 and 100 ms, respectively. AB Analyst TF Software (AB SCIEX, Framingham, USA) was used to collect data. The collision energy (CE) and collision energy superposition (CES) of each compound were optimized.

The Thermo Fisher LTQ-Orbitrap Electrostatic Mass Spectrometer was used in ESI negative ion mode and multistage scanning mode. The optimized parameters were as follows: capillary temperature, 320 °C; ion source heater, 300 °C; source voltage, 3.6 kV; sheath gas (N_2_), 35 arbitrary units; aux gas, 10 arbitrary units; sweep gas flow, 0 arbitrary units; S-Lens RF level (%): 60%; CE, 30 eV; and CES, 10 eV. The experiments were performed in negative mode, scanning from *m*/*z* 100 to 1000.

### 3.5. Data Preprocessing Analysis

The data collected via UHPLC-Q-TOF-MS/MS were processed using Peak View 1.2 software (AB SCIEX, version 1.2.0.3) from AB Sciex Company. The empirical molecular formulae were deduced from Peakview by comparing the theoretical masses of molecular ions and/or adductions with the determined values based on the following error limits: mass accuracy, < 5 ppm; retention time, < 5.0%; and isotope abundance, < 10%. The multistage mass spectrometry data collected via ESI-LTQ-Orbitrap-MS^n^ were processed using Thermo Xcalibur 2.2 (Thermo Fisher Scientific).

## 4. Conclusions

In this work, 15 PFs from *Artocarpus* plants served as standards and were divided into five types according to their structural characteristics with respect to the position and existence of prenyl substitution in the flavone skeleton. A data acquisition strategy based on the combination of Q-TOF-MS and LTQ-Orbitrap-MS^n^ was used to obtain both [M − H]^−^ and product ion information. Fragmentation rules for the five groups were summarized, and possible fragmentation pathways were proposed. These fragmentation rules were successfully exploited to qualitatively analyze the PFs from *A. heterophyllus*, a well-known *Artocarpus* plant; a total of 47 PFs were identified from the extract of this plant. These results indicate that the developed analytical method could be used as a rapid, effective technique for the qualitative characterization of PFs in *Artocarpus* plants and in other PF-accumulating medicinal herbs.

## Figures and Tables

**Figure 1 molecules-24-04591-f001:**
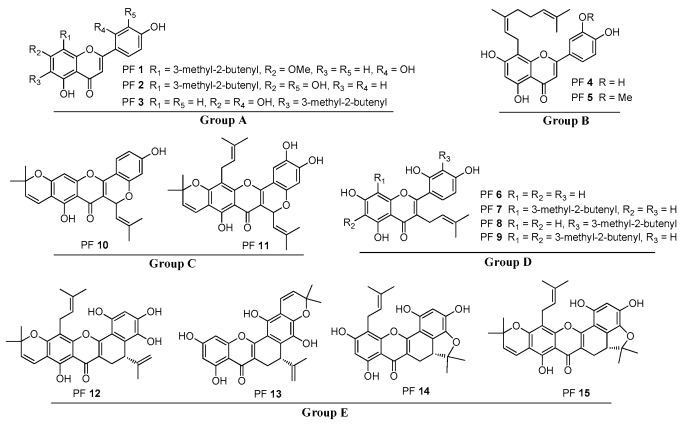
Structures of the five groups of prenylated flavonoids (PFs) selected as model molecules from *Artocarpus* plants.

**Figure 2 molecules-24-04591-f002:**
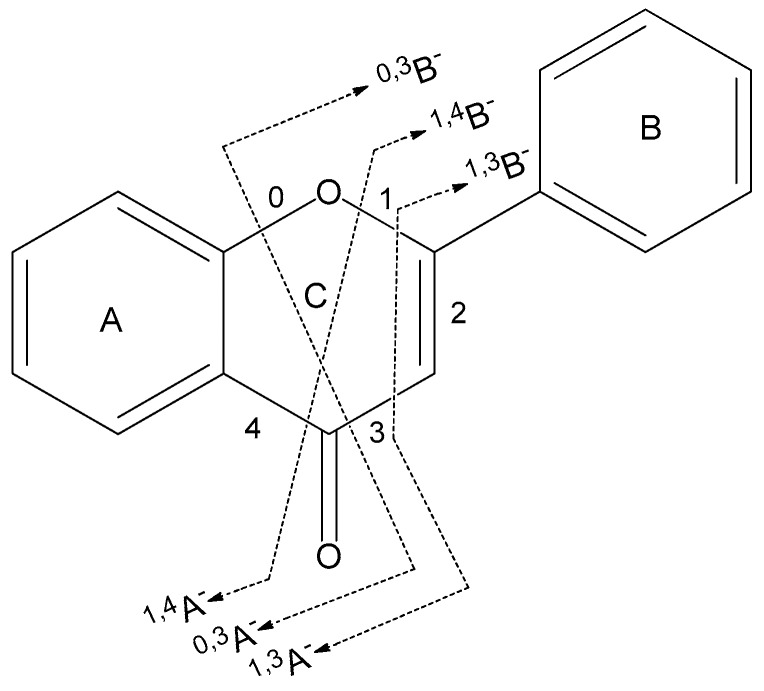
Nomenclature of the different C ring cleavages [21] occurring in the present study, illustrated by a typical flavonoid skeleton.

**Figure 3 molecules-24-04591-f003:**
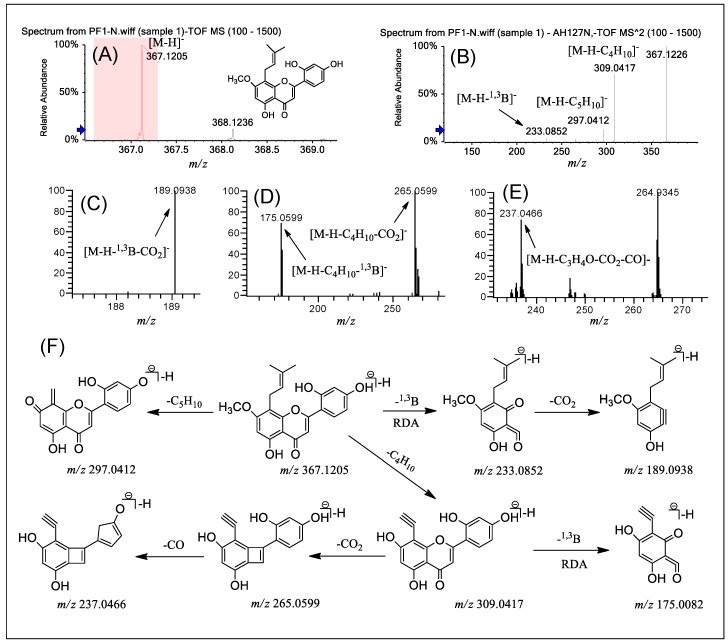
(−) ESI-Q-TOF-MS/MS and LTQ-Orbitrap-MS^n^ (*n* = 3–4) spectra of PF **1** in group A and the fragmentation pathways proposed for PF **1**. Q-TOF-MS (**A**), Q-TOF-MS^2^ (**B**), LTQ-Orbitrap-MS^3^ of the ions at *m*/*z* 233.0852 from *m*/*z* 367.1205 (**C**), LTQ-Orbitrap-MS^3^ of the ions at *m*/*z* 309.0417 from *m*/*z* 367.1205 (**D**), LTQ-Orbitrap-MS^4^ of the ions at *m*/*z* 265.0599 from *m*/*z* 309.0417 (**E**), and the proposed fragmentation pathways for PF **1** (**F**). RDA: retro Diels–Alder.

**Figure 4 molecules-24-04591-f004:**
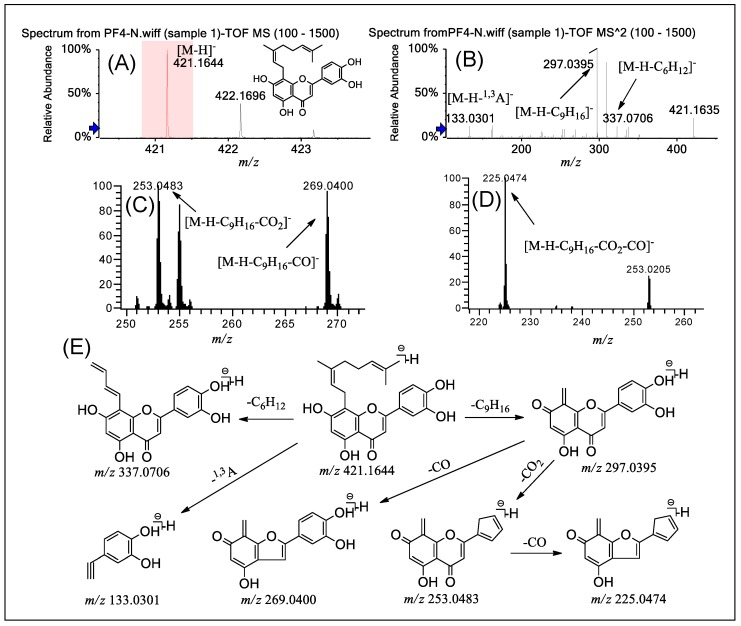
(−) ESI-Q-TOF-MS/MS and LTQ-Orbitrap-MS^n^ (*n* = 3–4) spectra of PF **4** in group B and the fragmentation pathways proposed for PF **4**. Q-TOF-MS (**A**), Q-TOF-MS^2^ (**B**), LTQ-Orbitrap-MS^3^ of the ions at *m*/*z* 297.0395 from *m*/*z* 421.1644 (**C**), LTQ-Orbitrap-MS^4^ of the ions at *m*/*z* 253.0483 from *m*/*z* 297.0395 (**D**), and the proposed fragmentation pathways for PF **4** (**E**).

**Figure 5 molecules-24-04591-f005:**
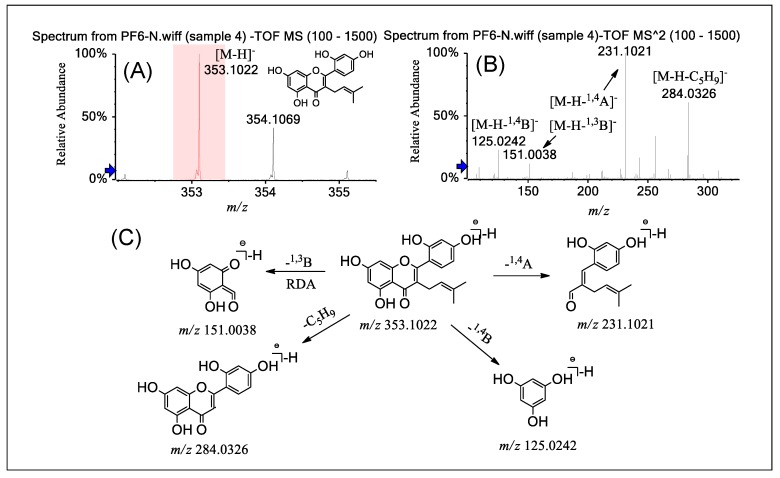
(−)ESI-Q-TOF-MS/MS of PF **6** in group C and the fragmentation pathways proposed for PF **6**. Q-TOF-MS (**A**), Q-TOF-MS^2^ (**B**), and the proposed fragmentation pathways for PF **6** (**C**). RDA: retro Diels–Alder.

**Figure 6 molecules-24-04591-f006:**
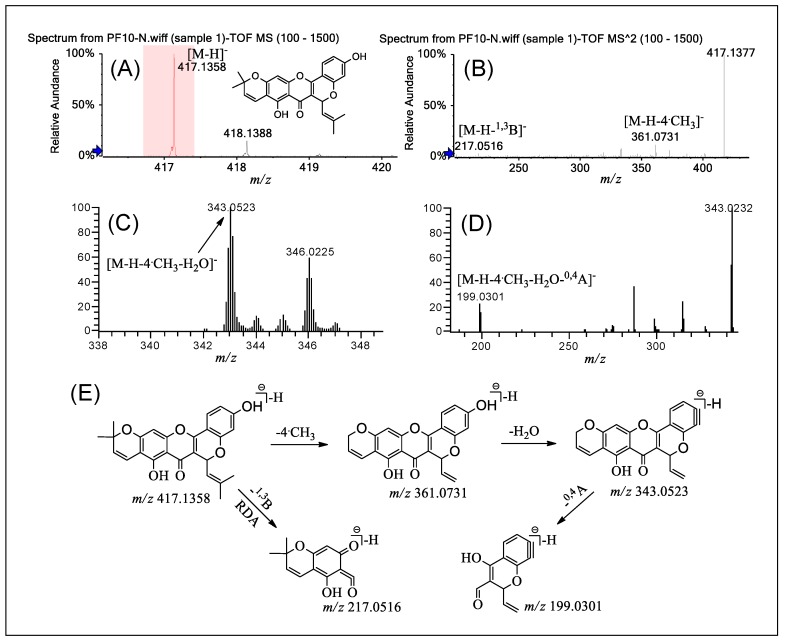
(−)ESI-Q-TOF-MS/MS and LTQ-Orbitrap-MS^n^ (*n* = 3–4) spectra of PF **10** in group B and the fragmentation pathways proposed for PF **10**. Q-TOF-MS (**A**), Q-TOF-MS^2^ (**B**), LTQ-Orbitrap-MS^3^ of the ions at *m*/*z* 361.0731 from *m*/*z* 417.1358 (**C**), LTQ-Orbitrap-MS^4^ of the ions at *m*/*z* 343.0523 from *m*/*z* 361.073 (**D**), and the proposed fragmentation pathways for PF 10 (**E**). RDA: retro Diels–Alder.

**Figure 7 molecules-24-04591-f007:**
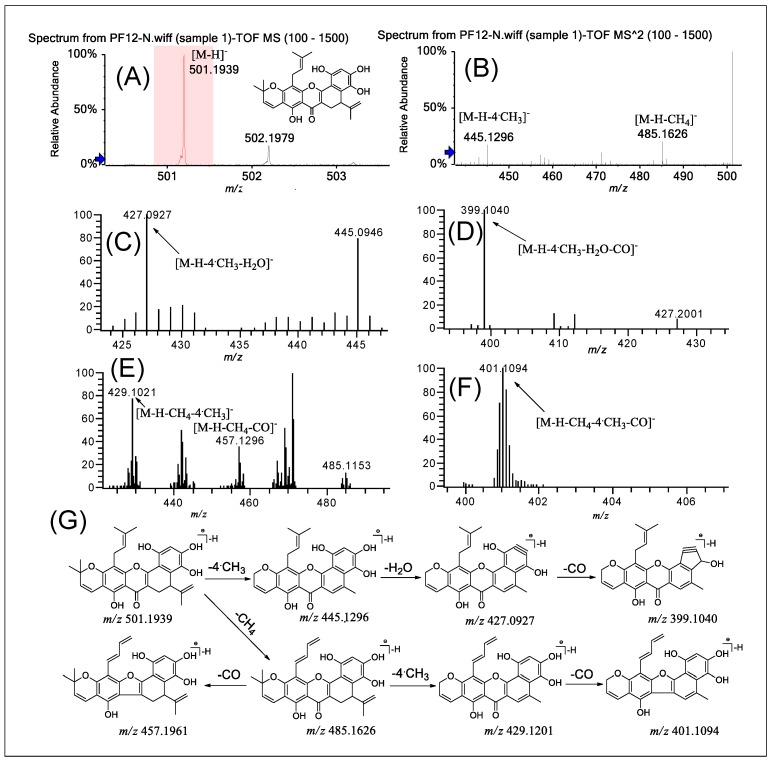
(−)ESI-Q-TOF-MS/MS and LTQ-Orbitrap-MS^n^ (*n* = 3–4) spectra of PF **12** in group B and the fragmentation pathways proposed for PF **12**. Q-TOF-MS (**A**), Q-TOF-MS^2^ (**B**), LTQ-Orbitrap-MS^3^ of the ions at *m*/*z* 445.1296 from *m*/*z* 501.1939 (**C**), LTQ-Orbitrap-MS^4^ of the ions at *m*/*z* 427.0927 from at *m*/*z* 445.1296 (**D**), LTQ-Orbitrap-MS^3^ of the ions at *m*/*z* 485.1631 from *m*/*z* 501.1939 (**E**), LTQ-Orbitrap-MS^4^ of the ions at *m*/*z* 429.1201 from *m*/*z* 485.1631 (**F**), and the proposed fragmentation pathways for PF **12** (**G**).

**Figure 8 molecules-24-04591-f008:**
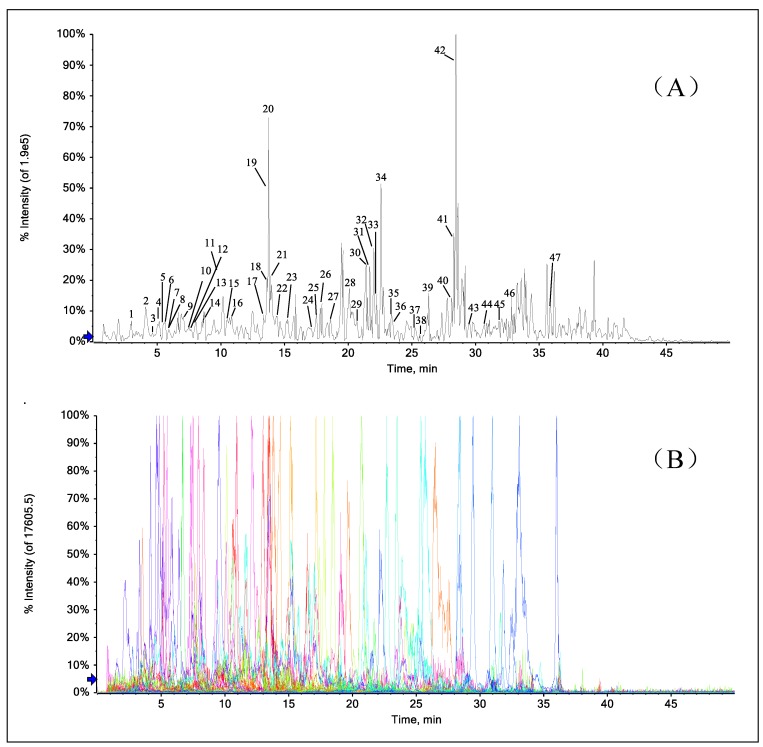
Total ion chromatogram (**A**) and extracted ion chromatogram (**B**) of prenylated flavonoids in the extract of *Artocarpus heterophyllus* Lam. roots.

**Table 1 molecules-24-04591-t001:** Precursor ions and main product ions observed in the electrospray ionization quadrupole time-of-flight mass spectrometry (ESI-Q-TOF-MS^n^) and linear trap quadrupole orbitrap mass spectrometry (LTQ-Orbitrap-MS^n^) spectra of the protonated PFs **1**–**15**.

Compounds	MS^n^	Precursor Ions	Product Ions *m*/*z* (Relative Abundance, %)
PF **1**	MS/MS	367 [M − H]^−^	175 (3.4); 233 (3.9); 265 (4.2); 297 (18.9); 309 (77.7); 337 (1.6); 352 (1.7); 367 (100.0)
	MS^3^	233	133 (20.4); 165 (61.7); 189 (23.7); 217 (100.0)
		309	175 (61.7); 265 (100.0); 291 (26.7)
	MS^4^	265	173 (100.0); 193 (36.4); 215 (62.1); 221 (75.1); 237 (70.1)
PF **2**	MS/MS	353 [M − H]^−^	133 (6.8); 297 (15.5); 309 (15.5); 353 (100.0)
	MS^3^	297	225 (20.1); 227 (21.4); 253 (97.5); 255 (83.4); 269 (100.0)
		309	265 (18.8); 267 (19.9); 309 (100.0)
	MS^4^	269	225 (53.4); 227 (35.2); 241 (75.7); 251 (100.0); 269 (39.3)
		267	211 (10.4); 223 (17.9); 239 (44.7); 252 (20.5); 267 (100.0)
PF **3**	MS/MS	353 [M − H]^−^	133 (2.0); 219 (9.8);309 (3.1); 353 (100.0)
	MS^3^	219	133 (80); 151 (39.4); 175 (100.0)
	MS^4^	175	65 (13.4); 133 (100.0)
PF **4**	MS/MS	421 [M − H]^−^	133 (12.7); 161 (8.9); 255(9.4); 269 (8.7); 297 (100.0); 323(12.4); 337 (12.0); 421 (21.9)
	MS^3^	297	163 (9.8); 225(22.8); 227 (22.1); 253(100.0); 255 (84.4); 269 (95.9)
	MS^4^	253	211 (10.7); 225 (100.0); 253 (22.9)
PF **5**	MS/MS	435 [M − H]^−^	133 (6.7); 269 (8.); 297(84.5); 309 (30.1); 311 (13.5); 323(21.1); 351 (100.0); 420 (35.4); 435 (32.8)
	MS^3^	297	163 (9.0); 225(20.9); 227 (21.0); 253(88.6); 255 (75.6); 269 (100.0)
		351	307 (9.2); 309 (19.1); 333 (24.0); 351 (100.0)
	MS^4^	253	211 (8.38); 225 (100.0); 253 (20.2)
PF **6**	MS/MS	353 [M − H]^−^	109 (16.7); 125 (30.6); 151 (24.8); 231 (100.0); 243 (13.2); 283 (11.6); 284 (62.0); 353 (50.5)
	MS^3^	231	145 (23.1); 147 (23.2); 149 (24.0); 151 (76.7); 163 (23.0); 187 (100.0); 189 (28.2)
		284	256 (100.0)
	MS^4^	256	122 (100.0); 212 (56.7); 227 (43.6); 228 (38.3)
PF **7**	MS/MS	421 [M − H]^−^	193 (8.91); 219 (1.9); 227 (1.8); 283 (2.4); 297 (7.7); 299 (20.9); 309 (38.8); 311 (3.1); 421 (100.0)
	MS^3^	193	125 (43.1); 149 (100.0); 151 (36.2)
		309	175 (92.2); 265 (100.0)
	MS^4^	149	107 (100.0)
		265	173 (59.8); 175 (44.4); 215 (74.5); 221 (76.7); 223 (100.0); 237 (56.2)
PF **8**	MS/MS	421 [M − H]^−^	193 (7.1); 293 (7.7); 299 (27.7); 309 (47.1); 421 (100.0)
	MS^3^	299	244 (90.0)
		309	175 (10.2); 244 (100.0); 256 (19.7); 299 (78.7)
	MS^4^	299	244 (100.0); 299 (58.0)
		244	123 (100.0); 149 (35.3); 150 (69.0); 203 (53.3); 215 (35.8); 216 (77.4); 229 (99.11)
PF **9**	MS/MS	489 [M − H]^−^	261 (6.1); 351 (6.0); 365 (10.3); 367 (100.0); 377 (75.9); 489 (90.4)
	MS^3^	367	257 (10.6); 269 (24.5); 312 (100.0); 367 (71.2)
		377	269 (10.1); 312 (31.5); 367 (100.0)
	MS^4^	312	257 (51.4); 269 (100.0)
PF **10**	MS/MS	417 [M − H]^−^	199 (2.3); 217 (3.0); 333 (7.3); 347 (2.4); 361 (11.7); 373 (6.29); 387 (2.8); 401 (3.3)
	MS^3^	361	319 (7.5); 334 (13.5); 343 (100.0); 346 (59.4)
	MS^4^	343	199 (22.1); 287 (36.2); 299 (10.0); 315 (24.2); 343 (100.0)
PF **11**	MS/MS	501 [M − H]^−^	285 (3.8); 429 (3.7); 457 (43.22); 473 (3.3); 501 (100.0)
	MS^3^	285	285 (5.2); 243 (22.0); 241 (100.0); 217 (8.5)
		457	402 (20.1); 439 (100.0); 442 (10); 457 (18.7)
	MS^4^	241	133 (67.5); 157 (32.2); 173 (77.2); 193 (26.6); 197 (100.0); 213 (32.5)
		439	357 (58.70); 383 (21.73); 395 (53.51); 411 (100.0); 424 (20.26)
	MS^5^	411	329 (35.9); 341 (87.8); 354 (73.8); 356 (87.3); 367 (54.4); 383 (76.9); 393 (58.3); 396 (100.0); 411 (22.5)
PF **12**	MS/MS	501 [M − H]^−^	415 (8); 417 (5.0); 429 (7.3); 445 (14.8); 457 (4.2); 471 (13.0); 483 (4.4); 501 (100.0)
	MS^3^	445	355 (25.3); 373 (38.8); 399 (39.0); 403 (26.7); 417 (100.0); 427 (94.2); 445 (74.6)
		485	429 (100.0); 442 (60.3); 457 (29.9); 469 (46.5); 471 (85.7)
	MS^4^	427	355 (53.9); 383 (10.5); 399 (100.0); 427 (34.5)
		429	401 (100.0); 411 (31.5); 414 (20.5)
	MS^5^	399	343 (52.5); 355 (100.0); 356 (41.6)
		401	357 (88.5); 373 (100.0); 383 (26.6); 386 (70.6); 401 (19.3)
PF **13**	MS/MS	433 [M − H]^−^	373 (11.3); 389 (37.0); 390 (63.2); 405 (25.3); 417 (11.2); 433 (100.0)
	MS^3^	389	361 (15.2); 372 (100.0); 373 (34.3); 375 (17.5); 389 (27.9)
		405	335 (17.4); 350 (32.9); 361 (27.9); 360 (30.4); 372 (21.9); 389 (15.1); 405 (100.0)
	MS^4^	372	326 (34.9); 327 (35.9); 343 (71.7); 371 (100.0)
		350	309 (21.6); 335 (100.0); 349 (22.2)
	MS^5^	343	299 (13.9); 343 (100.0)
		335	276 (16.0); 291 (68.1); 308 (25.6); 317 (34.2); 335 (100.0)
	MS^6^	299	263 (14.0); 276 (100.0); 291 (16.7)
		291	263 (14.0); 276 (100.0); 291 (16.7)
PF **14**	MS/MS	435 [M − H]^−^	349 (34.2); 363 (5.0); 377 (20.7); 379 (12.4); 391 (12.4); 393 (6.3); 419 (6.3); 435 (100.0)
	MS^3^	349	277 (21.4); 305 (15.0); 321 (32.9); 349 (100.0)
		391	337 (29.5); 349 (100.0); 363 (23.9); 377 (17.8)
	MS^4^	321	264 (100.0); 276 (40.7); 279 (62.5); 293 (90.0)
		337	309 (100.0)
	MS^5^	309	237 (21.1); 263 (32.6); 365 (23.3); 279 (27.0); 281 (70.3); 291 (100.0); 309 (31.4)
PF **15**	MS/MS	501 [M − H]^−^	417 (5.0); 429 (5.5); 443 (5.8); 445 (12.2); 457 (7.3); 471 (7.7); 485 (14.8); 501 (100.0)
	MS^3^	445	355 (27.4); 373 (41.3); 399 (38.9); 417 (99.6); 427 (100.0); 445 (80.8)
		485	429 (100.0); 442 (60.8); 457 (34.2); 469 (47.9); 471 (88.0)
	MS^4^	427	355 (26.4); 371 (56.0); 385 (27.2); 399 (100.0)
		429	357 (26.3); 374 (19.6); 401 (100.0); 411 (29.9)
	MS^5^	399	355 (100.0); 384 (18.3)
		401	357 (91.0); 373 (100.0); 383 (32.7); 385 (48.8); 401 (21.8)

**Table 2 molecules-24-04591-t002:** The 47 prenylated flavonoids identified from the extract of *Artocarpus heterophyllus* Lam. roots.

Peak No.	Compounds	RT (min)	Molecular Formula	[M − H]^−^ (*m/z*)	Error (Δppm)	Product Ions (*m/z*)	Error (Δppm)	Fragment Ions Identified
1	6-(3-Methylbutyl-2-enyl)apigenin [23]	3.49	C_20_H_18_O_5_	337.10903	2.6	219.0681	8.2	[M – H − ^1,3^B]^−^
						281.0473	6.4	[M − H − C_4_H_8_]^−^
2	Artonin K [24]	4.14	C_21_H_18_O_7_	381.09924	3.3	337.0738	5.9	[M – H − C_2_H_4_O]^−^
						365.0687	5.5	[M – H − CH_4_]^−^
3	Albanin A [25]	4.66	C_20_H_18_O_6_	353.10459	4.3	125.0246	4.8	[M – H − ^1,4^B]^−^
						231.0668	2.2	[M – H − ^1,4^A]^−^
4	14-Hydroxyartonin E [26]	4.87	C_25_H_24_O_8_	451.14188	4.5	393.0999	4.8	[M – H − C_3_H_6_O]^−^
						433.1289	0.9	[M − H − H_2_O]^−^
5	Artoindonesianin P [27]	5.22	C_22_H_20_O_7_	395.11548	4.7	337.0736	5.3	[M – H − C_3_H_6_O]^−^
						351.0515	1.4	[M – H − 3·CH_3_]^−^
6	Artonin J [28]	5.47	C_25_H_24_O_7_	435.14663	3.9	379.0839	4.2	[M – H − C_4_H_8_]^−^
						365.0681	3.8	[M – H − C_5_H_10_]^−^
7	Artelastoxanthone [18]	6.67	C_25_H_22_O_7_	433.12925	−0.1	405.1399	−1.2	[M – H − CO]^−^
						389.1017	3.6	[M – H − CO_2_]^−^
8	Artobiloxanthone [29]	6.68	C_25_H_22_O_7_	433.13224	6.8	405.1364	4.9	[M – H − CO]^−^
						403.0850	6.7	[M – H − 2CH_3_]^−^
9	Artonin E [28]	7.23	C_25_H_24_O_7_	435.14663	3.9	379.0841	4.7	[M – H − C_4_H_8_]^−^
						351.0887	3.7	[M – H − C_5_H_4_O]^−^
10	Cyclocommunol [30]	7.31	C_20_H_16_O_6_	351.08818	2.2	199.0777	6.0	[M – H − ^1,3^A]^−^
						295.0259	3.7	[M – H − C_4_H_8_]^−^
11	Artoindonesianin Q [4]	7.47	C_22_H_22_O_7_	397.12891	−0.9	137.0247	−2.2	[M – H − ^1,4^B]^−^
						313.0345	2.9	[M – H − CH_3_ − C_5_H_9_]^−^
						311.0195	0.6	[M – H − OCH_3_ − C_4_H_7_]^−^
						365.1021	−2.2	[M – H − OCH_3_]^−^
12	Artoindonesianin R [31]	7.49	C_22_H_22_O_7_	397.13136	5.1	313.0357	1.0	[M – H − C_6_H_12_]^−^
						137.0242	−1.5	[M – H − ^1,4^B]^−^
13	Artonol E [32]	7.91	C_26_H_24_O_7_	447.14548	1.2	415.1155	−7.7	[M – H − OCH_4_]^−^
						391.0831	2.0	[M – H − C_4_H_8_]^−^
14	Artoindonesianin T [31]	8.31	C_22_H_20_O_7_	395.11548	4.7	339.0518	2.4	[M – H − C_4_H_8_]^−^
						365.0655	−3.3	[M – H − 2CH_3_]^−^
15	Cycloartobiloxanthone [27]	10.48	C_25_H_22_O_7_	433.13134	4.7	391.0824	0.3	[M – H − C_3_H_6_]^−^
						415.1177	−2.4	[M – H − H_2_O]^−^
16	Artoindonesianin B [33]	10.92	C_26_H_28_O_8_	467.17322	4.4	261.0758	−3.8	[M – H − ^1,4^A]^−^
						451.1399	0.2	[M – H − CH_4_]^−^
17	Artocarpetin A [34]	13.07	C_21_H_20_O_6_	367.11769	−2.8	233.0829	−4.3	[M – H − ^1,3^B]^−^
						297.0410	−1.7	[M – H − C_5_H_10_]^−^
						309.0763	1.6	[M – H − C_3_H_4_O]^−^
18	Heteroflavanone C [35]	13.44	C_23_H_26_O_7_	413.16078	0.5	247.0975	−0.4	[M – H − C_9_H_10_O_3_]^−^
						235.0982	2.6	[M – H − ^1,2^B]^−^
19	Artocarpesin [36]	13.48	C_20_H_18_O_5_	337.10903	2.6	133.0294	−0.8	[M – H − ^1,3^A]^−^
						281.0477	7.8	[M – H − C_4_H_8_]^−^
20	6-(3-Methylbut-2-enyl) Apigenin [18]	13.5	C_20_H_18_O_5_	337.10809	−0.2	281.0464	−3.2	[M – H − C_4_H_8_]^−^
						117.0349	−2.6	[M – H − ^1,3^A]^−^
21	Styracifolin D [4]	13.88	C_30_H_32_O_8_	519.20258	0.3	461.1583	5.0	[M – H − C_3_H_6_O]^−^
						491.2075	0.0	[M – H − CO]^−^
						501.1919	0.0	[M – H − H_2_O]^−^
22	Cycloartocarpesin [37]	14.28	C_20_H_16_O_5_	335.09274	0.7	133.0301	4.5	[M – H − ^1,3^A]^−^
						305.0462	2.3	[M – H − 2CH_3_]^−^
23	Artonin U [38]	15.17	C_21_H_20_O_5_	351.12462	2.4	281.0469	5.0	[M – H − C_5_H_10_]^−^
24	Norartocarpin [39]	17.16	C_25_H_26_O_6_	421.16567	0	227.0725	4.8	[M – H − ^1,4^A]^−^
						309.0391	−4.5	[M – H − C_8_H_16_]^−^
						365.1009	-6.0	[M – H − C_4_H_8_]^−^
25	Mulberrochromene [40]	17.50	C_25_H_24_O_6_	419.15040	0.9	199.0781	8.0	[M – H − ^1,3^A]^−^
						349.0726	2.3	[M – H − C_5_H_10_]^−^
26	Dihydroisocycloartomunin [31]	17.79	C_26_H_26_O_7_	449.16080	0.5	191.0715	0.5	[M – H − ^1,4^B]^−^
						365.0694	7.4	[M – H − C_6_H_12_]^−^
27	Morusin [41]	18.48	C_25_H_24_O_6_	419.15040	0.9	191.0728	7.3	[M – H − ^1,4^B]^−^
						217.0515	4.1	[M – H − ^1,3^B]^−^
						227.0725	4.8	[M – H − ^1,4^A]^−^
28	Heterophyllin [17]	20.55	C_30_H_32_O_7_	503.20842	2.1	217.0879	4.1	[M – H − ^1,3^A]^−^
						243.0662	−0.4	[M – H − ^1,4^A]^−^
						259.1351	4.2	[M – H − ^1,4^B]^−^
						285.1112	5.6	[M – H − ^1,3^B]^−^
29	Artocarpetin B [42]	20.72	C_22_H_22_O_6_	381.13518	2.2	233.0806	−5.6	[M – H − ^1,3^B]^−^
						311.0549	−3.9	[M – H − C_5_H_10_]^−^
30	Artelastofuran [43]	21.07	C_30_H_34_O_7_	505.22363	0.9	277.1455	3.6	[M – H − ^1,4^B]^−^
						285.1147	5.3	[M – H − ^0,3^B]^−^
31	5’-Hydroxycudraflavone A [17]	21.81	C_25_H_22_O_7_	433.12925	−0.1	389.1372	5.7	[M – H − CO_2_]^−^
						415.1192	−1.2	[M – H − H_2_O]^−^
						217.0516	−4.6	[M – H − ^1,3^A]^−^
						377.0663	1.1	[M – H − C_3_H_4_O]^−^
32	Artonin F [44]	22.04	C_30_H_30_O_7_	501.19231	0.9	429.1343	−0.2	[M – H − C_4_H_8_O]^−^
						473.1945	−5.3	[M – H − CO]^−^
33	Isocycloheterophyllin [45]	22.23	C_30_H_30_O_7_	501.19290	−2.0	445.1267	−2.2	[M – H − CH_4_]^−^
						485.1615	−1.9	[M – H − C_3_H_4_O]^−^
						443.1511	−2.5	[M – H − C_3_H_6_O]^−^
34	Artoindonesianin I [46]	22.70	C_30_H_34_O_7_	505.22363	0.9	285.1137	1.8	[M – H − ^0,3^B]^−^
						447.1806	−1.6	[M – H − C_3_H_6_O]^−^
35	Artocarpin [47]	23.48	C_26_H_28_O_6_	435.18167	0.8	207.1039	5.8	[M – H − ^1,4^B]^−^
						233.0818	−0.4	[M – H − ^1,3^B]^−^
36	Cannflavin C [20]	23.59	C_26_H_28_O_6_	435.18022	−2.5	351.0878	−1.1	[M – H − C_6_H_12_]^−^
						323.0939	−4.3	[M – H − C_6_H_7_ − OCH_3_]^−^
						313.1452	−2.2	[M – H − C_9_H_14_]^−^
37	Artonin H [44]	25.10	C_30_H_34_O_7_	505.22363	0.9	191.0728	7.3	[M – H − ^1,4^B]^−^
						217.0512	2.8	[M – H − ^1,3^B]^−^
38	Artonin S [38]	25.36	C_26_H_28_O_7_	451.17621	0.0	393.1357	3.3	[M – H − C_3_H_6_O]^−^
						421.1311	4.3	[M – H − 2CH_3_]^−^
39	Artonin G [44]	26.21	C_30_H_32_O_7_	503.20862	2.2	447.1422	−6.0	[M – H − C_4_H_8_]^−^
						475.2103	−4.8	[M – H − CO]^−^
40	Artonin A [48]	28.15	C_30_H_30_O_7_	501.19231	0.9	457.1286	−1.5	[M – H − C_3_H_8_]^−^
						473.1950	−4.2	[M – H − CO]^−^
41	Artonin B [48]	28.39	C_30_H_30_O_7_	501.19230	0.8	485.1602	0.8	[M – H − CH_4_]^−^
						445.1296	−0.7	[M – H − C_3_H_4_O]^−^
42	Cudraflavone A [49]	28.41	C_25_H_22_O_6_	417.13475	0.9	217.0515	4.1	[M – H − ^1,3^B]^−^
						373.1072	−2.4	[M – H − C_2_H_4_O]^−^
43	Artoindonesianin G [46]	29.45	C_30_H_32_O_6_	487.21308	1.0	259.1351	4.2	[M – H − ^1,4^B]^−^
						285.1116	−5.6	[M – H − ^1,3^B]^−^
						365.1762	1.1	[M – H − C_7_H_6_O_2_]^−^
44	Cycloartocarpin [50]	30.97	C_26_H_26_O_6_	433.16608	1.0	199.0777	6.0	[M – H − ^1,3^A]^−^
						363.0878	1.1	[M – H − C_5_H_10_]^−^
						375.0878	1.1	[M – H − C_4_H_10_]^−^
45	Artoindonesianin H [46]	31.80	C_30_H_32_O_6_	487.21308	1.0	227.0731	7.5	[M – H − C_9_H_18_]^−^
						361.0731	3.6	[M – H − ^1,4^A]^−^
						473.1966	−0.8	[M – H − CO]^−^
46	Cycloheterophyllin [17]	33.05	C_30_H_30_O_7_	501.19231	0.9	285.1145	4.6	[M – H − ^1,3^B]^−^
						445.1281	−2.7	[M – H − C_4_H_8_]^−^
						473.1966	−0.8	[M – H − CO]^−^
47	Artelastochromene [43]	36.01	C_30_H_30_O_6_	485.19690	−0.1	285.1145	4.6	[M – H − ^1,3^B]^−^
						469.1659	0.4	[M – H − CH_4_]^−^

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
