# Peer review of "Characterization and Identification of Prenylated Flavonoids from Artocarpus heterophyllus Lam. Roots by Quadrupole Time-Of-Flight and Linear Trap Quadrupole Orbitrap Mass Spectrometry"

_molecules, 2019, doi:10.3390/molecules24244591_

Round 1
Reviewer 1 Report
This is an interesting paper and worth publishing, but currently there are a number of issues that need clarification and revision, mainly the fact the although the authors claim that they have identified 50 compouns with the proposed MS methodology, due to several duplicates, the number is lower. Besides, the English language needs improvement throughout the manuscript.
I recommend the revision/correction of the following aspects:
We recommend the use of “Artocarpus heterophyllus Lam.” in the title of the paper, as being more accurate. Moreover, because only the roots apparently were used as a source for the PFs, the title should reflect this (“Artocarpus heterophyllus Lam. roots”).
The RDA acronym used in the abstract should be explained (retro-Diels–Alder)
Page 1, line 37: “anti-respiratory burst activities” – this is meaningless. In fact, as per the reference used by the authors, it is an anti-inflammatory activity related to the inhibition of neutrophil respiratory burst. There is a big difference between “anti-respiratory burst” and “inhibition of neutrophil respiratory burst”, because the former would just suggest a stop of respiration, whereas the latter indicates an anti-inflammatory effect.
Page 1, line 44: “arbor tree” is pleonastic; one of the two words is sufficient.
Page 2, lines 47-48: “subdhing swelling and detoxicating” meaning is unclear in English, please try to use English equivalent words. “Subdhing” is not an English word, and “detoxicating” is too vague: against what kind of toxics it is used?
Lines 101-102: “PFs 1-3 mainly show the neutral loss of C4H8 or C5H10 …”. This statement is inconsistent with that on line 105, where it is written: “And common losses of other neutral molecules, such as C4H8, CO, and CO2, are also observed…”.
In figure 3, letter “G” should be replaced by “F”.
Line 255: “The extract of the A. heterophyllus roots was detected….” Detected is inappropriate here, the extract could be detected by the naked eye. Probably the authors meant that the extract was analyzed by UPLC-Q-TOF-MS/MS.
Lines 258-260: “The formulae of the fifty compounds were accurately assigned within mass error of 5 ppm, and the mass errors of characteristic fragment ions are all within 10 ppm.” It is not clear how the authors computed the 5 and 10 ppm errors.
Table 2, line 15: “Cylcoartobiloxanthone” should be corrected to “Cycloartobiloxanthone”.
Table 2, line 18: According to this PubChem record (https://pubchem.ncbi.nlm.nih.gov/compound/390520#section=Synonyms), “8-(gamma,gamma-dimethylallyl)-5,2',4'-trihydroxy-7-methoxyflavone” is a synonym for Artocarpetin A, whereas the authors of this draft paper shows it as different from Artocarpetin A. Please clarify this discrepancy against the available literature.
Table 2, line 25: According to this PubChem record (https://pubchem.ncbi.nlm.nih.gov/compound/5481958), Norartocarpin is a synonym for mulberrin. The authors, however, have reported mulberrin in Table 2 on line 3. If the two are indeed synonym, the authors should reconcile the two lines/compounds reported here.
Table 2, line 28: According to the reference cited by the authors, the name of the compound should be “morusin”, not “mosin”. The latter does not seem to exist.
Table 2, lines 29 and 31: the same substance (Heterophyllin) is reported twice, with different retention times. Considering the different RTs and fragmentation patterns, it is more likely the two are different compounds (as recognized by the authors, who claim that they identified 50 DIFFERENT compounds). The should check the data again and clarify this inconsistency.
Table 2, lines 33, 34 and 35: we could not find any evidence that a 5'-Hydroxycudraflavone exists that is different from 5'-Hydroxycudraflavone A (e. PubChem only contains records about 5'-Hydroxycudraflavone A.). The authors should at least provide structural formulas for the two allow an informed reader to understand the differences. Moreover, compounds form lines 34 and 35 are one and same, it is not clear why it is tabulated twice.
Table 2, line 49: please correct the name to “cycloheterophyllin”.
Table 2, line 50: please correct the name to “artelastochromene”.
Because of the several duplicates in Table 2, the number of 50 PFs claimed in the abstract and throughout of the paper is inaccurate. The authors should deal with the duplicates and correct the number of PFs reported here.
Reviewer 2 Report
Corrections
Line 88 - Table 1. Main product ions observed in the ESI-Q-TOF-MSn and LTQ-Orbitrap-MSn spectra of the protonated PFs 1-15. [M-H]- is a deprotonated molecule.
Line 104 - Explain better, peak at mz 297 was not formed by retro Diels-Alder (RDA) cleavage from the 1,3-position of C-ring.
Line 323 EtOH elution fraction was concentrated to afford a residue. The dried residue was re-dissolved in 50.
These features could help to differentiate the compounds bearing a free prenoid moiety at the C-6- from those at the C-8 positions
Round 2
Reviewer 1 Report
Thank you for improving the manuscript and congratulations for your work!